# Ceramides as Mediators of Oxidative Stress and Inflammation in Cardiometabolic Disease

**DOI:** 10.3390/ijms23052719

**Published:** 2022-02-28

**Authors:** Melania Gaggini, Rudina Ndreu, Elena Michelucci, Silvia Rocchiccioli, Cristina Vassalle

**Affiliations:** 1Institute of Clinical Physiology, National Research Council, Via G. Moruzzi 1, 56124 Pisa, Italy; mgaggini@ifc.cnr.it (M.G.); rudina.ndreu@ifc.cnr.it (R.N.); emichelucci@ifc.cnr.it (E.M.); silvia.rocchiccioli@ifc.cnr.it (S.R.); 2Fondazione CNR-Regione Toscana G Monasterio, 56124 Pisa, Italy

**Keywords:** ceramides, cardiovascular disease, metabolic diseases, oxidative stress, inflammation

## Abstract

Ceramides, composed of a sphingosine and a fatty acid, are bioactive lipid molecules involved in many key cellular pathways (e.g., apoptosis, oxidative stress and inflammation). There is much evidence on the relationship between ceramide species and cardiometabolic disease, especially in relationship with the onset and development of diabetes and acute and chronic coronary artery disease. This review reports available evidence on ceramide structure and generation, and discusses their role in cardiometabolic disease, as well as current translational chances and difficulties for ceramide application in the cardiometabolic clinical settings.

## 1. Introduction

Ceramides are a family of bioactive lipid molecules, composed by a sphingosine and a fatty acid [1]. Their chemical structure varies according to the levels of complexity and heterogeneity, due to the differences in chain length and degree of saturation of fatty acids, presence of polar groups, or variable sugar groups giving different glycosphingolipids. These differences may confer specific variable pathophysiological functions. To note, six different fatty acyl selective ceramide synthases (CerS) are involved in the production of ceramides, further complicating the scenario, and suggesting highly selective modulation as a great challenge for innovative therapeutic tools [2]. In fact, specific enzyme modulation may increase or decrease generation of specific adverse or protective ceramide species in different conditions, providing the possibility to develop inhibitors and activators targeting specific ceramides. Specifically, inhibition of glucosylceramide synthase (GlcCer synthase), sphingomyelinase (SMase) or ceramidase, which may possibly favor apoptotic ceramides, may be of interest in cancer settings [3].

Nonetheless, there are numerous studies involving enzymes of ceramide metabolism in cancer, since different ceramide species seem to be involved in several cellular pathways with different functions, possibly characterizing different pathophysiological states during cancer onset and development [4]. Thus, research based on the evaluation of activity and regulatory mechanisms of the enzymes implicated in ceramide metabolism is an important developing research area aiming to identify new therapeutic tools or to improve conventional therapies.

Moreover, inhibition of serine palmitoyltransferase (SPT) or ceramide synthases (CerS) may be of interest in inflammatory conditions as it lowers the amount of the inflammatory ceramides [5]. In the clinical context, there is important evidence on the association between ceramide species and cardiometabolic disease, especially in relationships with the onset and development of type 2 diabetes and acute and chronic coronary artery disease. Underlying mechanisms explaining these relationships involve many critical cellular pathways, including apoptosis, oxidative stress and inflammatory responses. The aim of this review is to go into more detail on the various aspects of ceramide metabolism and to discuss possible translational relevance of ceramide application in cardiometabolic diseases (Box 1).

Box 1The terms coronary artery disease (CAD), cardiovascular disease (CVD), and cardiometabolic disease (CMD) are closely interconnected and often used interchangeably. In the following box, definition and similarities/differences of these terms are reported.
Cardiovascular disease (CVD) is a general word used to indicate all conditions affecting the heart or blood vessels, but generally associated with the term atherosclerosis, which is characterized by fat deposition inside the vessel wall over time, narrowing the lumen and with development of lesions (plaques) and giving an increased risk of ischemic eventsCoronary artery disease (CAD) refers to onset and development of atherosclerotic plaque in arteries that supply blood to the heart.Cardiometabolic disease (CMD) included a group of common closely related and preventable conditions (atherosclerosis and coronary artery disease, stroke, diabetes, insulin resistance and non-alcoholic fatty liver disease).


## 2. Ceramide Structure and Production

Ceramides are sphingolipids composed of a sphingosine backbone N-acylated coupled to variable acyl side chains through an amide linkage. CerS are central enzymes required for the acylation reaction. So far, six different CerS (Cers1–6) have been described, differing by tissue location and substrate specificity, which produce different species of ceramides [2]. In particular, each CerS exhibits characteristic substrate preference toward acyl-CoAs carrying different chain-lengths (CerS1, C18; CerS2, C22 and C24; CerS4, C20, C22 and C24; and CerS5 and CerS6, C16), whereas only CerS3 exhibits broad substrate specificity toward medium- to long-chain fatty acyl-CoAs [6]. These ceramides can be further metabolized through the addition of different head groups to generate the complete arsenal of sphingolipids (e.g., sphingomyelins and glycosphingolipids). Different structures in distinct ceramide species, conferred by fatty acid chain length and saturation, may give specific variable (even opposite) biological effects. In particular, it has been suggested that shorter chains exert more adverse biological effects, whereas very long–chain ceramides may have more protective effects [7] (short-chain, ≤C18; medium-chain, C18–C22; long-chain, ≥C22). For example, CerS6 has an antiapoptotic role in endoplasmic reticulum (ER) stress response and produces mainly C16-ceramide, while CerS1 generates C18-ceramide mediating insulin resistance in skeletal muscle [2,8,9,10]. Important intracellular compartment of ceramide metabolism are mitochondria, which contain a variety of sphingolipids, including ceramides, and have their own subset of sphingolipid-generating and -degrading enzymes [10,11].

Ceramide biosynthesis can occur through three main distinct pathways (each characterized by several key enzymes; see Table 1): (1) de novo, (2) sphingomyelin hydrolysis, or (3) sphingolipid salvage [12].

(1) De novo synthesis takes place in the cytosolic layer of endoplasmic reticulum involving SPT, a heterodimer formed by 2 subunits, which is the rate-limiting enzyme of this pathway, catalysing the condensation of palmitate and serine to give 3-keto-dihydrosphingosine. Then, 3-keto-dihydrosphingosine reductase produces sphinganine, and then, dihydroceramides are generated under the action of different isoforms of ceramide synthase, through the incorporation of acyl-CoA of different chain lengths. Finally, dihydroceramide desaturase introduces a double bond in position 4–5 trans of dihydroceramide to form ceramides. This pathway is the major source of ceramides and is induced in inflammation and hypoxia. Once synthesized, ceramide is transported into the Golgi apparatus to generate other sphingolipids. In the Golgi apparatus, ceramides can be further metabolized to complex sphingolipids (e.g., sphingomyelin, glucosylceramides and gangliosides) and could be also phosphorylated by the ceramide kinase (CerK) to generate ceramide-1-phosphate (C1P), able to modulate insulin signaling in muscle cells and stimulate the incorporation of glucose by the glucose transporter type 4 Glut4 [13,14]. Interestingly, C1P is highly related to cardiovascular processes by stimulating migration, proliferation, angiogenesis, cell survival, and metabolism [11];

(2) Sphingomyelinase pathway, a catabolic way to produce ceramides and phosphocholine by the degradation of sphingomyelin through acid sphingomyelinase (ASM). This is a catabolic pathway, considered as a “fast” way to generate ceramides, that involves enzymes which hydrolyze sphingomyelin in the cell membrane producing ceramides and phosphocholine;

(3) Salvage pathways occur in endo-lysosomes where ceramides are produced from glycosphingolipids through ceramide synthase.

## 3. Ceramides and Cardiometabolic Risk

### 3.1. Ceramides, Cardiovascular Risk and Disease

Circulating ceramides originate in the liver, a major site of de novo ceramide synthesis [15]. Lipoprotein transport ceramides in blood; in particular, the very-low-density lipoprotein (VLDL) and low-density lipoprotein (LDL, about 80%), whereas the remaining percentage is carried by albumin and high density lipoproteins (HDL) [16]. The heart is the main site of metabolism of circulating lipids within triglyceride-rich (TG-rich) lipoproteins, providing substrates for ceramide production. The role of ceramides in acute coronary events involves the ability of ceramide to promote oxidized low-density lipoprotein infiltration into the vessel wall, facilitating monocyte adhesion, atherosclerotic plaque generation, and rendering the lipid-rich core of the plaque larger and more prone to rupture [17]. Accordingly, two recent imaging studies evidenced the localization of ceramides with the thin fibrotic plaques with necrotic core [18,19]. In the Dallas Heart Study, 1557 participants without type 2 diabetes underwent measurements of metabolic biomarkers, fat depots by magnetic resonance imaging-MRI and plasma ceramides measurement, while the diabetes onset was assessed after 7 years. In this study, the total cholesterol was associated with all ceramides, while higher triacylglycerols and lower HDL resulted as associated only with saturated fatty acid chain ceramides, and visceral adipose tissue was positively associated with saturated fatty acid ceramides and inversely associated with polyunsaturated fatty acid ceramides [17]. However, none of the ceramides were independently associated with incident prediabetes or type 2 diabetes after adjustment for clinical factors [20].

In the general population, serum ceramides emerged as novel biomarkers of atherosclerosis and overall cardiovascular risk, resulting closely associated with lower aerobic capacity [21], with female gender, aging, cardiometabolic risk factors, as overweight/obesity, hypertension, insulin resistance and type 2 diabetes, dyslipidemia [22,23] and predictive of adverse cardiovascular outcomes [24,25,26,27].

Ceramides increased with aging, as reported by different studies [22,23,28,29,30]. In particular, experimental data suggested that the differential elevation in sphingolipid metabolic enzymes enhanced ceramide production in different organs of aged animals [31]. Other data obtained through manipulation of sphingolipid metabolism using pharmacological and genetic tools in caenorhabditis elegans suggested that accumulation and remodeling of specific ceramides (e.g., dC18:1-C24:1), gangliosides (e.g., GM1-C24:1), and sphingomyelins (e.g., dC18:1-C18:1) are critical factors to determine development rate and lifespan in the nematode model [32].

In humans, the Strong Heart Family Study, including a large cohort of adult subjects (2145 subjects; 41% men), evidenced higher levels of ceramide species in older participants [33]. Although it is not completely clear if there are differences in ceramide levels between male and female subjects, some data reported positive association between age and ceramides (e.g., Cer (d18:1/24:0) and Cer (d18:1/24:1)), especially in postmenopausal women [22,23]. Importantly, estradiol appears to inversely correlate with Cer (d18:1/24:1) in women, while the incubation with estradiol (10 nM, 24 h), decreases ceramide accumulation in cancer cells expressing estrogen receptors, giving strength to the hypothesis of a close relationship between ceramide levels and estradiol [29].

Interestingly, a recent review reported available genetic studies, underlying the role of some genetic variants involved in ceramide biosynthesis as significant determinants of heritable circulating ceramide levels, and their relationship with certain cardiometabolic-related traits [30].

Recent data obtained in a general Czech population of 461 adults evidenced Cer (d18:1/14:0) changes associated to altered blood pressure, total cholesterol and fasting blood glucose, identifying this lipid specie as a possible biomarker of cardiometabolic risk stratification in subjects without cardiovascular disease [34]. Nonetheless, recent data from The Cardiovascular Health Study evidenced that the prognostic capacity (risk of death) evaluated in a large community-based cohort of elderly adults (*n* = 4612; ≥65 years) followed in the years 1992–2015, differs according to the length of their acylated saturated fatty acid. In fact, high concentrations of Cer carrying fatty acid 16:0 were related with an increased risk of mortality, whereas high concentrations of ceramides carrying longer fatty acids were associated with a decreased risk of mortality [33]. Among other lipid metabolites and lipidomic patterns, Cer16:0 was found significantly associated with heart failure (HF) risk in 2 different cohorts (a case-control study of 331 incident HF cases and 507 controls, nested within the PREDIMED-Prevención con Dieta Mediterránea study), suggesting that ceramides have potential additive biomarkers to assess HF risk [35].

These findings were in agreement with those reported by the Framingham Heart Study (FHS), where high circulating ceramide Cer 24:0/C16:0 were inversely associated with incident cardiovascular disease and with incident HF [36]. Moreover, the Cardiovascular Health Study (CHS) evidenced that lower circulating concentrations of fatty acids with 24 carbons and no unsaturated bonds, was associated with a higher risk of HF [37].

Thus, ceramides appear as new promising biomarkers to assess novel atherosclerosis and overall cardiovascular risk even in the general population. In particular, Cer18:1/C16:0, Cer18:1/C18:0 and Cer18:1/C24:1 can efficaciously predict cardiovascular events in asymptomatic individuals [24].

Ceramides also resulted associated with an increased incident of HF in a large general population, and Cer16:0/C24:0 in particular are related to preclinical left ventricular dysfunction, remodelling and HF [38]. A very recent review evidenced how high cardiac levels of total ceramides may characterize HF. In addition, increased incident HF, overall cardiovascular disease mortality and all-cause mortality were also associated with higher Cer16:0/C24:0 (or lower Cer24:0/C16:0) [39].

Importantly, ceramides may retain even better predictivity over traditional lipids, identifying residual risk in coronary artery disease patients with low LDL-C [40,41]. Distinct ceramides progressively increased from control subjects to patients with stable angina, unstable angina, acute myocardial infarction (AMI), and correlates with the percentage of stenosis (as an index of disease severity) [42,43,44,45]. Moreover, recent data indicated a positive and independent relationship between plasma ceramides and the presence of plaque rupture (respect to plaque erosion, by optical coherence tomography examination for culprit plaque) in AMI patients [46]. Interestingly, ceramides retained significant prognostic value, being independently associated with major adverse cardiovascular events (AMI, percutaneous intervention, coronary artery bypass, stroke, or death within 4 years) in patients with and without coronary artery disease [47]. In particular, Cer (d18:0/20:0) was found to be associated with calcified plaque volume (quantified by using computed tomography coronary angiograms) in subjects with and without cardiovascular risk factors, aged 41–75 years [48]. There is evidence that cardiac ceramides accumulate in the failing myocardium, and increased levels are detectable in blood [27]. By contrast, inhibition of de novo ceramide synthesis reduces cardiac remodeling, improving myocardial systolic function. In vitro experiments revealed that changes in ceramide synthesis are linked to hypoxia (oxidative stress) and inflammation. Thus, increased de novo ceramide synthesis appears closely involved in cardiac remodeling and dysfunction, whereas its inhibition could be very important in the treatment and prevention of the failing heart [38]. Nonetheless, the relationship between ceramides and cardiovascular disease and events is determined by their acyl composition and saturation. For example, ceramides containing the C16 and C18 acyl chains (C16:0 and C18:0) appear particularly dangerous, whereas those containing very long chains (e.g., C24 or C24:1) appear to be either protective or benign. In particular, one group of researchers identified three ceramide species (C16:0, C18:0, and C24:1) more promising in terms of association with stable and acute coronary artery disease and metabolic abnormalities (e.g., insulin resistance). Therefore, since the three latter ceramides are considered good predictors of adverse cardiovascular events, a ceramide-based score was proposed, it is named CERT1 and included these species and their ratio to C24:0 for normalization [40,41]. We recently evidenced that ceramide species and ratios in the CERT1 score resulted associated with cardiovascular risk, inflammation and disease severity in acute myocardial infarction [49]. Thus, their evaluation may help to better understand CV pathobiology and suggests these new biomarkers as possible risk predictors and pharmacological targets in AMI patients [50]. This score was then updated to the CERT2 score, which also includes phosphatidylcholines (PC) (composed by one ceramide/ceramide ratio, two ceramide/PC ratios and a single PC), that efficaciously predicts CV events in patients [51]. Table 2 reports in detail components of the CERT1 and CERT2 scores. Instead, the Sphingolipid Inclusive Score (SIC), which includes several minor lipid molecules, has been recently proposed and seems to be more efficacious in patient stratification than traditional cardiovascular disease biomarkers (including conventional lipids, e.g., LDL cholesterol and triglycerides) [52]. Prognostic power of CERT and CERT2 was recently assessed in a large population (*n* = 999) of Austrian cardiovascular disease patients, followed for up to 13 years, where both scores resulted powerful predictors of cardiovascular events, cardiovascular mortality, and overall mortality [53].

A further evolution led to a sphingolipid-based risk score for coronary artery disease (the sphingolipid-inclusive SIC score), combining some lipid species to a base conventional risk factor model (11 conventional risk factors: age, sex, total cholesterol, HDL-C, current smoking, nature of prior acute coronary syndrome, revascularization, diabetes history, stroke history, history of hypertension, and randomized treatment allocation), which improved the prediction of cardiovascular outcomes, suggesting the potential of plasma lipidomic profiles as biomarkers for cardiovascular risk stratification in secondary prevention [54]. Moreover, the dScore was also proposed to estimate a 10-year absolute risk (scale 0–100) of developing diabetes, including not only ceramides- Cer(d18:1/18:0)/Cer(d18:1/16:0), but also anthropometric biomarkers (age, sex and body mass index) [55]. Now, all these new tools appear promising and useful beyond traditional lipids, particularly the CERT1, the most simple to calculate and interpret (advantage that cannot be neglected), although for all scores the capacity to stratify patients in the different clinical cardiovascular settings need to be further validated in further large-scale cohort studies. The identification of few relevant ceramides/scores (as C16:0, C18:0, and C24:1 and their ratios to C24:0, that predicted risk and severity of cardiovascular disease) may thus represent the first step to target the most promising lipid species for a reliable and easier use in CV diagnosis, risk stratification, and treatment. Importantly, both pharmacological (e.g., statins) and life-style intervention (diet, aerobic exercise) may modulate ceramide profile [1,56,57,58]. Interestingly, co-administration of tyrosol (a dietary phenolic compound) and white wine decreased levels of three ceramide ratios (Cer C16:0/Cer C24:0, Cer C18:0/Cer C24:0, and Cer C24:1/Cer C24:0) in parallel to improved endothelial function (assessed by augmentation index, an indirect measure of arterial stiffness, traditional lipids, D-dimer, homocysteine, C-reactive protein), in subjects at high cardiovascular disease risk [59].

Moreover, recent findings investigate the potential of using simple dietary interventions to improve biomarkers related to cardiovascular risk through the gut microbiome composition and its metabolic functions. Interestingly, it was observed that omega-3 and fiber supplementation decreased plasma ceramides (d18:1/16:0, d18:0/24:0, and d18:1/24:1), which were associated with the reduction in the amount of Colinsella and increases in Bifidobacteriuim and Coprococcus 3 and short-chain fatty acid [60].

Circadian-rhythm disruption is emerging as another risk factor for obesity and metabolic diseases which, although self-sustained, may be affected in synchrony and oscillation amplitudes by environmental factors, such as time of feeding. Accordingly, experimental data imposing time-restricted feeding (TRF) to Drosophila melanogaster flies is able to counteract obesity-induced dysmetabolism and improves muscle performance (e.g., suppressing intramuscular fat deposits, phospho-AKT level, mitochondrial alterations, and markers of insulin resistance), involving sphingosine kinase 2 (Sk2), which modulates ceramides production [61].

Moreover, sustaining daily rhythms in feeding and fasting under time-restricted feeding (TRF) without reducing caloric intake can reduce the risk of metabolic disease among overweight or obese humans [62,63].

Now, the effect of TRF on muscle or blood ceramides requires further interpretation. Although it may be a research area of great interest, as TRF is, between the different dietary strategies, it may represent an effective dietary intervention to face metabolic-related disease.

Aerobic exercise has been shown to modulate circulating ceramide levels [64]. In particular, cardiorespiratory fitness (CRF), a significant biomarker of health status, the improvement of which is related to a reduced incidence of non-communicable diseases and mortality, resulted as inversely associated to ceramide levels, after analysis in a systematic review reporting the association between CRF and metabolites measured in human tissues and body fluids, and including 22 studies [65]. In this context, additional therapies directed towards the inhibition of ceramide pathways (e.g., myriocin-inhibitor of serine palmitoyl-CoA transferase, an enzyme involved in up-regulation in de novo synthesis of ceramide), ceramide inhibitors (e.g., Fumonisin B1, fungin FTY720, etc.), as well as inhibition of the downstream inflammatory pathway (e.g., methotrexate, which targets CERS6) are under study [66,67].

### 3.2. Ceramides, Insulin-Resistance and Type 2 Diabetes

Several studies reported the relationship between high levels of circulating and accumulation of ceramides are associated with impaired insulin signaling and glucose transport, insulin resistance, inflammation, oxidative stress, and beta-cell apoptosis that are the main independent risk factors of cardiometabolic disease [67,68]. As mentioned above, the biological activity of sphingolipids is affected by the different types of saturated fatty acid content, as well as their association with insulin resistance and type 2 diabetes. By the use of the hyperinsulinemic-euglycemic clamp technique, Haus et al. showed that insulin sensitivity resulted as lower in type 2 diabetes subjects, and that the concentrations of C18:0, C20:0, C24:1 and total ceramides were significantly higher compared to controls, while insulin sensitivity was inversely correlated with C18:0, C20:0, C24:1, C24:0, and total ceramides [69]. Moreover, in type 2 diabetic subjects, tumor necrosis factor (TNF)-α levels were increased and correlated with increased C18:1 and C18:0 ceramides [70]. A quantitative analysis in the plasma of individuals who will progress to diabetes up to 9 years before disease onset, revealed that specific long-chain fatty-acid-containing dihydroceramides were significantly elevated [71]. A recent study showed that circulating levels of ceramides containing acylated palmitic acid (Cer-16), stearic acid (Cer-18), arachidic (Cer-20), and behenic acid (Cer-22) were each associated with a higher risk of diabetes in the Cardiovascular Health Study (a large cohort study of cardiovascular disease among elderly adults) [72]. In the FINRISK 2002 study, four ceramides Cer(d18:1/16:0), Cer(d18:1/18:0), Cer(d18:1/24:0) and Cer(d18:1/24:1) were analyzed, and the most significant predictor both in univariate and multivariable analyses for incidence of type 2 diabetes was the Cer(d18:1/18:0)/Cer(d18:1/16:0) ratio [55]. In the PREDIMED trial, it was found that sphingomyelin d18:1/18:0 was associated with a reduced risk of incident type 2 diabetes [73]. In the Mayo Clinic Study of the Aging cohort, the associations of ceramides with prevalent type 2 diabetes at baseline and incident type 2 diabetes during median follow up of 6.2 years were examined, after adjusting for demographic and metabolic factors. The results showed that ceramides were associated with prevalent type 2 diabetes (C16:0, C18:0, C18:0/C16:0 ratio, C18:0/C24:0 ratio) and incident type 2 diabetes (C18:0, C18:0/C16:0 ratio), and this could be useful for primary and secondary prevention of type 2 diabetes [74]. Nonetheless, the population of this recent study predominantly included older white adults with a high burden metabolic disease and differed from other study populations of younger, healthier adults of diverse races and ethnicities [75,76]. Table 3 summarizes the ceramides’ profiles evaluated in the discussed T2DM and IR studies. Interestingly in this context, very recent data show that liraglutide (a GLP-1 receptor agonist) decreased ceramide levels, suggesting downstream effects of liraglutide on lipid metabolism that could benefit the cardiovascular profile in type 2 diabetes patients [77].

## 4. Ceramides and Molecular Mechanism in Cardiometabolic Disease

Ceramides have a distinct role in several metabolic and cardiometabolic diseases. In particular several ceramide species play a role in apoptosis by signalling from the plasma membrane at pro-apoptotic receptors that lead to the development of atherosclerotic plaque [78]. In subjects with insulin resistance, type 2 diabetes, non-alcoholic fatty liver diseases and obesity, plasma circulating ceramides were elevated and related to the major adverse cardiovascular events, including mortality and correlated with the increase of inflammatory cytokines [79,80]. Different metabolic derangements are associated with ceramides accumulation. In particular, we focused on insulin resistance and obesity, inflammation and oxidative stress.

✓Ceramide accumulation inhibits glucose uptake by serine/threonine kinase inhibition that is fundamental for insulin-stimulated glucose transporter GLUT4 translocation [81]. Moreover, elevated ceramides promote insulin-resistance by other different mechanisms (e.g., activating intracellular inflammatory processes and increasing cytokine production by macrophage through toll-like receptor-dependent and -independent mechanisms) [16]. Importantly, excessive saturated free fatty acids-FFA induce the accumulation of ceramide, which inhibit early steps in insulin signaling through the inhibition of Akt/PKB phosphorylation (a serine/threonine kinase involved in insulin-stimulated anabolic metabolism), indicating ceramides as critical intermediate linking saturated fats to the inhibition of insulin signaling [82]. Moreover, Akt inhibition blocks insulin signaling, and promotes excessive lipid storage by inhibiting hormone-sensitive lipase (HSL). CerS and the different isoforms (Cers1–6) are responsible for acylation reactions that produce different species of ceramides and differ by tissue location as well as substrate specificity [2]. In several studies, ceramide species, i.e., C16:0 and C18:3, in muscle were negatively correlated with insulin sensitivity [8], while serum C18:0 increased in obese type 2 diabetes subjects compare to their healthy controls [70]. In the murine knock-out model of CerS1, (the enzymes that synthesize ceramide C18:0), distinct ceramides were analyzed in the skeletal muscle. CerS1 is the predominant isoform in muscle and the C18:0 ceramides are the major ceramide subspecies found in this tissue [8]. CerS1 mRNA expression and C18:0 ceramide synthesis were increased in the skeletal muscle of high fat diet-fed obese mice, and the ablation of CerS1 in skeletal muscle of mice, improves glucose metabolism predominantly in liver by the increment of circulating levels of the fibroblast growth factor 21-FGF21 [9]. In a study performed in mice and humans, the RNA expression of CERS6 and C16:0 was higher in visceral and subcutaneous adipose tissue of obese humans, and the increment of CERS6 expression correlates with insulin resistance. In CerS6-deficient mice, there was a reduction in C16:0 ceramides and an improvement of glucose tolerance and the ablation protected from diet-induced obesity [9] (Figure 1). Yki-Järvinen et al. reported that in subjects with higher liver fat, the adipose tissue is inflamed, and the ceramides’ total amount increased independently from body mass index in subjects with normal liver fat content. In the same study, the expression in adipose tissue of the sphingomyelinases phosphodiesterase SMPD1 and SMPD3 were significantly greater in the high liver fat subjects vs. subjects with normal liver fat content [83].✓TNF-α is able to increase ceramide content in plasma and adipose tissue by the stimulating the expression of ceramide synthesis genes such as SPT, SMPD1 and SMPD3 [84,85]. Moreover, TNF-α increases ceramide generation leading to nuclear factor kappa B (NfκB) activation and inducing apoptosis [86]. The activation of TLR4 (a receptor involved in innate immune responses) by free fatty acids (FFA) and lipopolysaccharides (LPS) increased de novo synthesis of ceramides by activation of several enzymes for ceramides synthesis with the intervention of NF-κB, an obligate intermediate in these TLR-4 mediated effects on ceramide production.

Fibrinolysis is a physiological event aimed to prevent blood clots and dissolve fibrin clots. Increased levels of plasminogen activator inhibitor-1 (PAI-1), the primary blood inhibitor of tissue plasminogen activator (tPA), are related to the inhibition of plasmin degradation of fibrin, and to a decreased fibrinolytic capacity. In the brain, tPA is reported to regulate blood–brain barrier function, promote neurodegeneration, microglial activation, neurite outgrowth and regeneration [87]. Now, the use of recombinant tissue plasminogen activator recombinant tissue-type plasminogen activator (rTPA) represents the only effective pharmacological treatment for ischemic stroke [88]. Treatment of cultured human astrocytes with N-acetylsphingosine, that increase intracellular ceramide via activation of ceramide synthase or sphingomyelin hydrolysis, induced the release of t-PA and decreased the PAI-1 release [89]. Thus, a better kwnoledge of the connection between ceramides, tPa and PAI-1 may be particulary interesting for the possibility to target ceramides in order to modulate the tPA/PAI-1 relationship. Nonetheless, the variation of the ceramide profile seems to depend on the ischemia and to affect the outcome of the patients. In addition to the increase of total ceramide, experimental data from stroke models suggested that the levels of several long-chain and very-long-chain ceramides in brain cells are significantly increased after ischemic injury [90]. The accumulation of specific long-chain ceramides, including Cer(d18:1/16:0), Cer(d18:1/18:0), and Cer(d18:1/20:0) togheter with a sphingomyelin decrease has been observed in the gerbil hippocampus during the reperfusion phase [91]. Other data suggested that Cer(d18:1/14:0) and Cer(d18:1/16:0) values significantly increase and that their ceramide synthase isoform, CerS 5 appear upregulated in neurons after hypoxia/reoxygenation [92]. Importantly, following subarachnoid hemorrhage, patients with high concentrations of Cer(d18:1/18:0) in the cerebrospinal fluid presented poor outcomes [93]. Thus, collectively these results identify a novel role for ceramides in the pathogenesis of ischemia and postischemic processes and in their contribution to the prothrombotic phenotype in ischemia/reperfusion. Further studies are needed to investigate the potential of distinct ceramides and S1P as predictors of prognosis and as therapeutic targets, and to elucidate their regulatory reciprocal relationship with t-PA, and its inhibitor PAI-1.

As adipokine, glycoprotein PAI-1 is also able to increase the accumulation of ceramides in adipocytes and conversely ceramides can induce PAI-1 expression in adipocytes, revealing a bidirectional interplay between PAI-1 and ceramides in inflammation and metabolism of adipose tissue [94,95]. Accordingly, several studies showed that PAI-1 contributes to the worse of obesity, insulin resistance, type 2 diabetes, and atherothrombosis [96,97]. Many evidence reported a positive association between circulating ceramides and different inflammatory biomarkers (e.g., c-reactive protein) and cytokines [44,98]. Moreover, S-SMase activity (the enzyme which catalyzes the cleavage of sphingomyelinase to ceramides) seems to be mediated by inflammation [99].

✓Ceramides are physiological components of mitochondrial membranes, and they are necessary for normal mitochondrial function, but excess mitochondrial ceramide content has been shown to induce mitochondrial dysfunction and impair respiratory capacity by generation of reactive oxygen species and increase of oxidative stress, depletion of ATP and disruption the electron transport chain, apoptosis, and alteration of the permeability of the mitochondrial outer membrane [100]. In particular, the reciprocal relationship between ceramides and oxidative stress and the role of ceramides as second messengers involved in the induction of apoptosis is well known [101]. In fact, if the reactive oxygen species (e.g., H_2_O_2_) can increase ceramide generation, ceramides may directly act on mitochondria by the inhibition of isolated mitochondrial electron transport at complex III, leading to the increase of reactive oxygen species [102,103]. Figure 2 summarizes some mechanisms related to ceramides inflammation, oxidative stress and glucose uptake;✓Interestingly, a few studies elucidated some mechanistic details, evidencing the involvement of Sk2 in modulating ceramides production and inducing onset and development of atherosclerosis and other cardiac and skeletal muscle disorders, and suggesting SK2 potential significance as a new target of pharmacological interventions [104,105].

## 5. Translational Issues

Cardiometabolic disease remains the most prevalent cause of morbidity and mortality, despite the improvement of efficacious and impactful interventions on traditional risk factors. Thus, the need for new reliable biomarkers in this field remains high. Lipids have key roles in cardiometabolic disease, where distinct ceramide species and ratios have been proposed as critical determinants, due to their importance in pathways related to lipid and glucose metabolism, inflammation and oxidative stress. Interestingly, these molecules represent significant predictors for cardiometabolic events, and their prediction capacity is also relevant in patients under statins, suggesting ceramides as a potential index of residual cardiometabolic risk [41]. Moreover, since ceramides take part in many key pathophysiological processes (especially oxidative stress/inflammatory pathways) and are involved in the onset and development of cardiometabolic diseases, they could serve as new potentially therapeutic targets. Ceramide measurement in clinical practice is conceivable and forthcoming by a technical point of view, although currently not a standard practice [41]. It is true that currently these techniques may can be expensive and lack standardized measurements; moreover, the measurement is performed in a limited number of laboratories, as not all are equipped with spectrometry instrumentation for routine purposes, and require expertise trained personnel necessary to run such instrumentations (Table 4). Importantly, the extreme complexity in the structure of ceramides (e.g., chain-length) reflects differences in their biological properties, which may confer even opposite effects to different species. Thus, further knowledge is required to understand and utilize these molecules as diagnostic and predictor biomarkers, and targets of novel therapy. However, many improvements in this field have been made, for example, currently many isotope-labelled standards provide precise quantification and analytical stability (Table 4).

## 6. Future Directions

Measurement of the complex ceramide profile in large cohorts is currently necessary to evaluate and understand which species and ratios may be reliable biomarkers in different pathophysiological cardiometabolic conditions. Efforts towards this objective surely require close cooperation from biologists, biochemists and clinicians, combining different competencies in a multidisciplinary collaborative team. In fact, the identification of a restricted number of few relevant ceramides species and/or their ratio in panels/scores (also in combination with other circulating, clinical or instrumental risk biomarkers) may facilitate comprehension of results by clinicians and the translational passage in the routine clinical practice. Some attempts are in progress, such as the addition of ceramides to high-sensitivity troponin T and traditional factors, which have been proven to ameliorate diagnostic performance in order to identify acute coronary events in patients with chest pain.

Moreover, the combination of the CERT2 score (composed by one ceramide/ceramide ratio, two ceramide/phosphatidylcholine-PC ratios and a single PC) and high-sensitivity troponin T has significantly improve the prognostic capacity in coronary artery disease patients [51].

## 7. Conclusions

Ceramide evaluation could be useful to better understand underlying pathogenic mechanisms associated with the cardiometabolic disease (especially in association with oxidative stress/inflammatory processes), and could be exploited in the future as additional tools in the armamentarium against cardiometabolic disease, even beyond traditional available lipid biomarkers, in view of their appreciable clinical potential and reliable prognostic capacity.

## Figures and Tables

**Figure 1 ijms-23-02719-f001:**
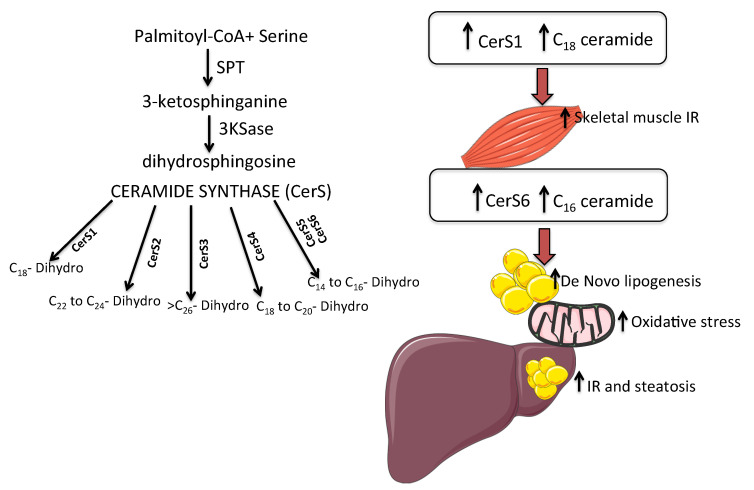
Ceramide synthases (CerSs) and the different isoforms (Cers1–6) are responsible for acylation reactions that produce different species of ceramides. In particular, CerS6 produces C16 ceramide that leads to insulin resistance (IR), de-novo lipogenesis and mitochondrial dysfunction. CerS1 mediated C18 ceramide that is related to skeletal muscle insulin resistance.

**Figure 2 ijms-23-02719-f002:**
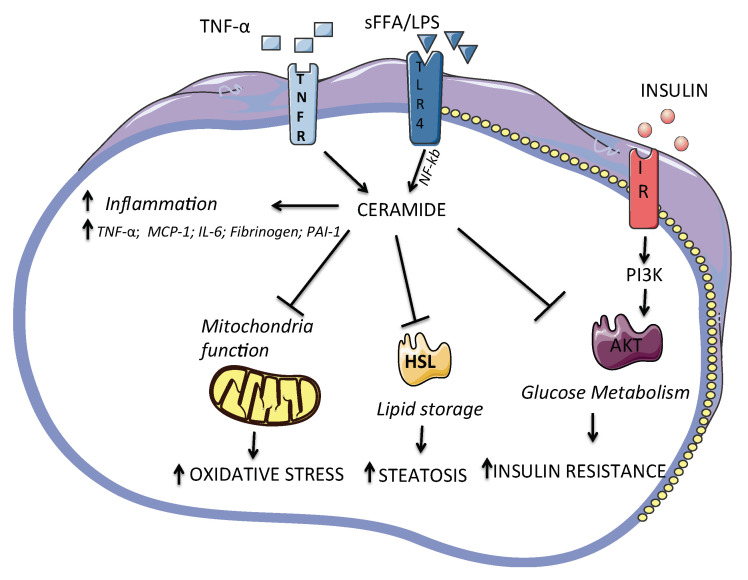
Ceramide accumulation leads to deleterious effects such as the inhibition of Akt to abrogate insulin signaling, promoting excessive lipid storage by inhibiting HSL (hormone-sensitive lipase), inhibits mitochondrial capacity leading to oxidative stress. Moreover, ceramides accumulation increments inflammation by the release of several pro-inflammatory cytokines. Akt: protein kinase B; PI3K: phosphatidylinositol-3-kinase; LPS: lipopolysaccharide; NF-κB: nuclear factor kappa-light-chain-enhancer of activated B cells; PAI-1: plasminogen activator inhibitor 1; sFFA: saturated fatty acids; TLR4: toll-like receptor 4; TNF-α: tumor necrosis factor alpha; IL-6: interleukine6; TNFR: tumor necrosis factor alpha receptor; MCP-1: monocyte chemoattractant protein-1; IR: insulin receptor.

**Table 1 ijms-23-02719-t001:** Key pathways of sphingolipid metabolism.

Pathways and Reactions of Ceramide Metabolism	Rate-Limiting Enzyme	Site of Production and Location	Other Main Involved Enzymes
De novo synthesis pathway	Serine palmitoyltransferase (SPT)	Endoplasmic reticulum	3-keto-sphingosine reductase Ceramide synthase (CerS) Dihydroceramide desaturase (Des1/DEGS1)
Sphingomyelinase pathway	Sphingomyelinases (Smases)	Plasma membrane; lysosomes	Sphingomyelin synthase (SMS)
Salvage pathway	Ceramide synthase (CerS)	Endo/lysosomes; mitochondria	Glucosylceramide synthase (GCS) Acid β-Glucosidase (β-GCase) Ceramidase
Ceramide kinase and ceramide 1 phosphate			Ceramide phosphate phosphatases (CPP)
Sphingosine kinase (SphK)/S1P phosphatase			Sphingosine kinase (SphK) Sphingosine 1-Phosphate Phosphatase (SPP) Lipid phosphate phosphatase (LPP)

**Table 2 ijms-23-02719-t002:** Components of the CERT1 and CERT2 scores.

Components of CERT1 Score	Components of CERT2 Score
Cer (d18:1/16:0)	Cer(d18:1/24:1)/(d18:1/24:0
Cer (d18:1/18:0)	Cer (d18:1/16:0)/PC 16:0/22:5
Cer(d18:1/24:1)	Cer (d18:1/18:0)/PC 14:0/22:6
Cer (d18:1/16:0)/(d18:1/24:0)	PC 16:0/16:0
Cer (d18:1/18:0)/(d18:1/24:0)	
Cer(d18:1/24:1)/(d18:1/24:0	

**Table 3 ijms-23-02719-t003:** Ceramides species evaluated in insulin-resistance and type 2 diabetes studies.

Ceramides Species	Study and/or Studied Subjects	Relation of Ceramides Levels and Related Disease	References
C18:0, C20:0, C24:1 and total	T2DM	Increased in plasma Insulin sensitivity decreased	Haus et al. [70]
C18:1 and C18:0	T2DM	Increased in plasma TNF-α increased	Haus et al. [70]
Cer-16, Cer-18, Cer-20, and Cer-22	Cardiovascular Health Study	Increased in plasma Risk of diabetes increased	Fretts et al. [72]
Cer(d18:1/16:0), Cer(d18:1/18:0), Cer(d18:1/24:0); Cer(d18:1/24:1) Cer(d18:1/18:0)/Cer(d18:1/16:0)	FINRISK study	Increased in plasma Incidence of type 2 diabetes increased	Hilvo et al. [55]
C16:0; C18:0, C18:0/C16:0 ratio, C18:0/C24:0 ratio	The Mayo Clinic Study of the Aging cohor	Increased in plasma Prevalence and incidence type 2 diabetes increased	Dugani et al. [74]
Dihydroceramide species Cer(d18:0)	Nine years before T2DM Diagnosis	Increased in plasma T2D predisposition	Wigger et al. [71]

**Table 4 ijms-23-02719-t004:** Main ceramide advantages and disadvantages for Translational and Clinical Applications.

Advantages	Disadvantages
isotope labelled standardsavailability of panel of ceramides and scores, which renders easier interpretation by clinicians	expensive testspectrometry instrumentation expensiveexpert and trained personnel neededdifficulties in result interpretationnot widespread in clinical laboratories
cardiovascular risk stratification in asymptomatic subjects and cardiometabolic patients	lack of reference ranges,lack of standardization,lack of control quality schemeslack of identification of key ceramides and panels/scores and the knowledge of their biological effects in different cardiometabolic conditionsevaluation of additive ceramide value over and beyond conventional biomarkers and CV scoresintra- and inter-biological variability to be assessed
targets of drugs and life-style therapy (e.g., anti-inflammatory agents)	identification of serum ceramide determinants needed (e.g., gender, age, circadian rhythm, and lifestyle)the extraction phase of the sample is operator-dependentpossible ionic suppression caused by the nature of the matrix
residual risk biomarkers in coronary artery disease patients

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
