# Peer review of "Ceramides as Mediators of Oxidative Stress and Inflammation in Cardiometabolic Disease"

_ijms, 2022, doi:10.3390/ijms23052719_

Round 1

Reviewer 1 Report

In this review, Gaggini et al provided available evidence on ceramide structure and generation and discusses their role in cardiometabolic disease, as well as actual translational changes and difficulties for ceramide application in cardiometabolic clinical settings. In general, this is a quite interesting and emerging field and most of the information provided is suitable for this publication. However, the authors need to add the following questions and need to include a few important references before considering this for publication.

  1. Circulating ceramides originate in the liver, and are transported to the bloodstream via lipoproteins. However, this section is just briefly mentioned and needs more expansion, including how they included cardiac diseases. The authors cited several population-based studies linking ceramide accumulation with the age.
  2. The authors have used the terms coronary artery disease (CAD), cardiovascular disease (CVD), and cardiometabolic disease (CMD) throughout the review and used CMD in the title. Each of the cardiac diseases is unique from each other and the authors need to clarify how frequently used these terms? Moreover, the authors need to outline CAD, CVD in the generation of CMD.
  3. In Figure 1, the author included skeletal muscle connection particularly in references to ceramides induced-insulin resistance, without talking too much about skeletal muscle involvement. However, the authors barely connected in the genesis of ceramides-induced CMD. Moreover, some mechanistic works are linking Sk2-induced ceramides production in the generation of cardiac and skeletal muscle disorders (Walls, S. M. et al, Cell report 2018 and Villanueva, J. E. et at Nature Communication, 2019).
  4. However, the author did not include these references and they need to discuss the relevance of these findings to this review (Walls, S. M. et al, Cell report 2018 and Villanueva, J. E. et at Nature Communication, 2019).
  5. Also, the authors included dietary reduction for inhibiting, ceramides-induced disorders without too much justification/evidence. In one of the above-mentioned manuscripts, the authors used feeding/fasting rhythms using TRF (to suppress mutant Sk2 the skeletal muscle dysfunctions by suppressing abnormal lipid-infiltration to and suppression insulin-resistance (Villanueva, J. E. et at Nature Communication, 2019).
  6. The figures are very poor quality and most of them shoed big black boxes within the figures. The authors need to improve figures quality and clarity.

Author Response

In this review, Gaggini et al provided available evidence on ceramide structure and generation and discusses their role in cardiometabolic disease, as well as actual translational changes and difficulties for ceramide application in cardiometabolic clinical settings. In general, this is a quite interesting and emerging field and most of the information provided is suitable for this publication. However, the authors need to add the following questions and need to include a few important references before considering this for publication. The authors cited several population-based studies linking ceramide accumulation with age.

  1. Circulating ceramides originate in the liver, and are transported to the bloodstream via lipoproteins. However, this section is just briefly mentioned and needs more expansion, including how they included cardiac diseases.

This part was further deepened at the beginning of 3.1 part

  1. The authors have used the terms coronary artery disease (CAD), cardiovascular disease (CVD), and cardiometabolic disease (CMD) throughout the review and used CMD in the title. Each of the cardiac diseases is unique from each other and the authors need to clarify how frequently used these terms? Moreover, the authors need to outline CAD, CVD in the generation of CMD.

Thank you for your clarification, you are right. These terms are surely interconnected, and as such used interchangeably. However, for an accurate definition and to evidence similarities/differences of these terms we added the following box in the text:

BOX. 1 The terms coronary artery disease (CAD), cardiovascular disease (CVD), and cardiometabolic disease (CMD) are closely interconnected and as such used interchangeably. However, find in the box definition and similarities/differences of these terms

Cardiovascular disease (CVD) is a general word used to indicate all conditions affecting the heart or blood vessels, but generally associated with the term atherosclerosis, which is characterized by fat deposition inside the vessel wall over time, narrowing the lumen and with development of lesions (plaques) and giving an increased risk of ischemic events.

Coronary artery disease (CAD) refers to onset and development of atherosclerotic plaque in arteries that supply blood to the heart.

Cardiometabolic disease (CMD) included a group of common closely related and preventable conditions (atherosclerosis and coronary artery disease, stroke, diabetes, insulin resistance and non-alcoholic fatty liver disease).

  1. In Figure 1, the author included skeletal muscle connection particularly in references to ceramides induced-insulin resistance, without talking too much about skeletal muscle involvement. However, the authors barely connected in the genesis of ceramides-induced CMD. Moreover, some mechanistic works are linking Sk2-induced ceramides production in the generation of cardiac and skeletal muscle disorders (Walls, S. M. et al, Cell report 2018 and Villanueva, J. E. et at Nature Communication, 2019).

This point better discussed and references added according to reviewer suggestions as follow:

“                                         

Circadian-rhythm disruption is emerging as another risk factor for obesity and metabolic diseases, that although self-sustained, may be affected in synchrony and oscillation amplitudes by environmental factors, such as time of feeding. Accordingly, experimental data imposing time-restricted feeding (TRF) to Drosophila melanogaster flies is able to counteract obesity-induced dysmetabolism and improves muscle performance (e.g. suppressing intramuscular fat deposits, phospho-AKT level, mitochondrial alterations, and markers of insulin resistance), involving sphingosine kinase 2 (Sk2), which modulates ceramides production.Villanueva JE, Livelo C, Trujillo AS, Chandran S, Woodworth B, Andrade L, Le HD, Manor U, Panda S, Melkani GC. Time-restricted feeding restores muscle function in Drosophila models of obesity and circadian-rhythm disruption. Nat Commun. 2019 Jun 20;10(1):2700.Moreover, sustaining daily rhythms in feeding and fasting under time-restricted feeding (TRF) without reducing caloric intake can reduce the risk of metabolic disease among overweight or obese humans (Sutton EF, Beyl R, Early KS, Cefalu WT, Ravussin E, Peterson CM. Early Time-Restricted Feeding Improves Insulin Sensitivity, Blood Pressure, and Oxidative Stress Even without Weight Loss in Men with Prediabetes. Cell Metab. 2018 Jun 5;27(6):1212-1221.e3.

Gabel K, Hoddy KK, Haggerty N, Song J, Kroeger CM, Trepanowski JF, Panda S, Varady KA. Effects of 8-hour time restricted feeding on body weight and metabolic disease risk factors in obese adults: A pilot study. Nutr Healthy Aging. 2018 Jun 15;4(4):345-353.

At now, the effect of TRF on muscle or blood ceramides require further deepening, although it may be a reserch area of great interest, as TRF, between the different dietary strategies, may represent an effective dietary intervention to face metabolic-related disease.”

  1. However, the author did not include these references and they need to discuss the relevance of these findings to this review (Walls, S. M. et al, Cell report 2018 and Villanueva, J. E. et at Nature Communication, 2019).

See the point above

  1. Also, the authors included dietary reduction for inhibiting, ceramides-induced disorders without too much justification/evidence. In one of the above-mentioned manuscripts, the authors used feeding/fasting rhythms using TRF (to suppress mutant Sk2 the skeletal muscle dysfunctions by suppressing abnormal lipid-infiltration to and suppression insulin-resistance (Villanueva, J. E. et at Nature Communication, 2019).

This point was better deepened according to reviewer suggestions as follow:

“Moreover, sustaining daily rhythms in feeding and fasting under time-restricted feeding (TRF) without reducing caloric intake can reduce the risk of metabolic disease among overweight or obese humans (Sutton EF, Beyl R, Early KS, Cefalu WT, Ravussin E, Peterson CM. Early Time-Restricted Feeding Improves Insulin Sensitivity, Blood Pressure, and Oxidative Stress Even without Weight Loss in Men with Prediabetes. Cell Metab. 2018 Jun 5;27(6):1212-1221.e3. Gabel K, Hoddy KK, Haggerty N, Song J, Kroeger CM, Trepanowski JF, Panda S, Varady KA. Effects of 8-hour time restricted feeding on body weight and metabolic disease risk factors in obese adults: A pilot study. Nutr Healthy Aging. 2018 Jun 15;4(4):345-353.

At now, the effect of TRF on muscle or blood ceramides require further deepening, although it may be a reserch area of great interest, as TRF, between the different dietary strategies, may represent an effective dietary intervention to face metabolic-related disease.”

  1. The figures are very poor quality and most of them shoed big black boxes within the figures. The authors need to improve figures quality and clarity.

Figures were downloaded as separated files

Reviewer 2 Report

The article reviews the literature on the involvement of ceramides in inflammation and oxidative stress in diseases encompassed as cardiometabolic. The work, however, has multiple flaws.

Major:

  • The metabolism of sphingolipids is not well explained. For example,
    or not all the enzymes involved are explained, naming only a few of them.
    - It is stipulated that sphingomyelinases are found only in the plasma membrane, however, there are secreted and acidic in lysosomes.
    - CerS are also located in mitochondria, which is related to oxidative stress and cell death, however, the author didn't explain it
    - The importance of the different isoforms of CerS, which give rise to Cer of different chains, is not explained. An important point for the subsequent explanation of blood sphingolipid profiling.
    - 3 enzymes involved in cancer are named, however, there are a variety of studies that implicate the diversity of enzymes in cancer.
    - The section on the different scores in sphingolipidomics is of great interest. A summary table should be made and the differences explained in depth.
    - It is explained that Cer inhibits the incorporation of glucose by Glut4, however, it is not mentioned that C1P stimulates it. This species is not even named throughout the work.
  • - There is a lack of information about the molecular mechanisms implicated in oxidative stress and inflammation due to ceramides.
    - None of the images is seen correctly so it is impossible to assess them.
    - Table 2 is not half visible.
    - Numerous grammatical errors are found.

Author Response

The article reviews the literature on the involvement of ceramides in inflammation and oxidative stress in diseases encompassed as cardiometabolic. The work, however, has multiple flaws.

Major:

  • The metabolism of sphingolipids is not well explained. For example,
    or not all the enzymes involved are explained, naming only a few of them.

We added other main enzymes involved in ceramide metabolism in Table 1. However, please consider that we wanted to summarize this part, which was more focused in other recent previous publications (also by our group: Gaggini M, Pingitore A, Vassalle C. Plasma Ceramides Pathophysiology, Measurements, Challenges, and Opportunities. Metabolites. 2021 Oct 21;11(11):719. )

- It is stipulated that sphingomyelinases are found only in the plasma membrane, however, there are secreted and acidic in lysosomes.

We better specify this point (see table 1 and text) according to reviewer suggestions

- CerS are also located in mitochondria, which is related to oxidative stress and cell death, however, the author didn't explain it

We better explained this point as follow:

“TNF-α is able to increase ceramides content in plasma and adipose tissue by the stimulating the expression of ceramide synthesis genes such as serine palmitoyltransferase (SPT), SMPD1 and SMPD3[74,75]. Moreover, TNF-α increases ceramide generation leading to NF-Æ™B activation and inducing apoptosis [Kitajima I., Soejima Y., Takasaki I., Beppu H., Tokioka T., Maruyama I. Ceramide-induced nuclear translocation of NF-κB is a potential mediator of the apoptotic response to TNF-α in murine clonal osteoblasts. Bone. 1996;19:263–270. Demarchi F., Bertoli C., Greer P.A., Schneider C. Ceramide triggers an NF-κB-dependent survival pathway through calpain. Cell Death Differ. 2005;12:512–522].”

and

“Moreover ceramides could induce apoptosis by its ability to form protein-permeable channels in mitochondrial membranes, since in mitochondria there are the enzymes responsible for ceramide synthesis and hydrolysis.((Bionda et al., 2004;)......................... De novo synthesis take place in the cytosolic layer of endoplasmic reticulum and mitochondria “

- The importance of the different isoforms of CerS, which give rise to Cer of different chains, is not explained. An important point for the subsequent explanation of blood sphingolipid profiling. We better specified this important point as follow:

In particular each CerSs exhibits a characteristic fatty acyl-CoA preference (CerS1, C18; CerS2, C22 and C24; CerS4, C20, C22 and C24; and CerS5 and CerS6, C16), only CerS3 exhibits broad substrate specificity toward medium- to long-chain fatty acyl-CoAs “               

“(short-chain≤C18; medium-chain, C18–C22; and long-chain, ≥C C22). For example CerS6 has an antiapoptotic role in ER stress responses and produces mainly C16-ceramide while CerS1 generates C18-ceramide mediating insulin resitance in skeletal muscle.[2;71;72]. “

- 3 enzymes involved in cancer are named, however, there are a variety of studies that implicate the diversity of enzymes in cancer.

This point was better explained according to reviewer suggestions as follow:

“Specifically, inhibition of glucosylceramide synthase (GlcCer synthase), sphingomyelinase (SMase) or ceramidase, which may possibly favor apoptotic ceramides, may be of interest in cancer settings [3]. Nonetheless, there are a variety of studies that implicate the diversity of enzymes in cancer, which is a major topic of study in this research area. Distinct ceramide species appear involved in different cellular pathways and functions, and likely characterized different pathophysiological states in cancer onset and development (Gomez-Larrauri A, Das Adhikari U, Aramburu-Nuñez M, Custodia A, Ouro A. Ceramide Metabolism Enzymes-Therapeutic Targets against Cancer. Medicina (Kaunas). 2021 Jul 19;57(7):729.). Thus, evaluation of their levels and the regulation of the enzymes implicated in their metabolism is a very interesting area to identify new therapeutic tools or improve conventional therapies.

- The section on the different scores in sphingolipidomics is of great interest. A summary table should be made and the differences explained in depth.

The need for further validation of these biomarkers was more explained in the text, and an additional table reporting components of CERT1 and CERT2 scores was added.

Moreover,      advantages and disadvantages for translational and clinical applications common to ceramide evaluation and scores are reported in Table 3.                       

- It is explained that Cer inhibits the incorporation of glucose by Glut4, however, it is not mentioned that C1P stimulates it. This species is not even named throughout the work.

we better explained this aspect adding in the text following part

Synthesized ceramide is then transported into the Golgi apparatus to generate other sphingolipids by two types of ceramide transport. In the Golgi apparatus, ceramides can be further metabolized to complex sphingolipids (e.g., sphingomyelin, glucosylceramides and gangliosides) and could be also phosphorylated by the ceramide kinase (CerK), for generate ceramide-1-phosphate (C1P) able to modulate insulin signaling in muscle cells and stimulating the incorporation of glucose by Glut4 (Ceramide-1-phosphate in cell survival and inflammatory signaling. Adv Exp Med Biol. (2010)

  • - There is a lack of information about the molecular mechanisms implicated in oxidative stress and inflammation due to ceramides.

We added:

“The reciprocal relationship between ceramides and oxidative stress and the role of ceramides as ceramide as a second messenger involved in the induction of apoptosis it is well known (Andrieu-Abadie N, Gouazé V, Salvayre R, Levade T. Ceramide in apoptosis signaling: relationship with oxidative stress. Free Radic Biol Med. 2001 Sep 15;31(6):717-28.). In fact, if the reactive oxygen species (e.g. H2O2) can increase ceramide generation, ceramides may directly act on mitochondria by the inhibition of isolated mitochondrial electron transport at complex III, leading to increase of reactive oxygen species [Andrieu-Abadie N, and Gaggini M, Sabatino L, Vassalle C. Conventional and innovative methods to assess oxidative stress biomarkers in the clinical cardiovascular setting. Biotechniques. 2020 Apr;68(4):223-231.

and …………………….“TNF-α is able to increase ceramides content in plasma and adipose tissue by the stimulating the expression of ceramide synthesis genes such as serine palmitoyltransferase (SPT), SMPD1 and SMPD3[74,75]. Moreover, TNF-α increases ceramide generation leading to NF-Æ™B activation and inducing apoptosis [Kitajima I., Soejima Y., Takasaki I., Beppu H., Tokioka T., Maruyama I. Ceramide-induced nuclear translocation of NF-κB is a potential mediator of the apoptotic response to TNF-α in murine clonal osteoblasts. Bone. 1996;19:263–270. Demarchi F., Bertoli C., Greer P.A., Schneider C. Ceramide triggers an NF-κB-dependent survival pathway through calpain. Cell Death Differ. 2005;12:512–522].2

  • None of the images is seen correctly so it is impossible to assess

We downloaded them as separate files

  • Table 2 is not half visible.

We add in text as figure

  • Numerous grammatical errors are found.

The text was revised by a certificated English reviewer in the new version

Round 2

Reviewer 2 Report

The authors have done a substantial job of improving the article. However, there are still a variety of grammatical errors. Apart from this point, the work has weak points:

Major:
- Line 91. "could be also phosphorylated by the ceramide kinase (CerK), to generate ceramide-1-phosphate (C1P) able to modulate insulin signaling in muscle cells and stimulating the incorporation of glucose by the glucose transporter type 4 Glut4"

C1P is not only a molecule that stimulates metabolism, something that the authors have mentioned referring to a review and not to the original article. C1P is a molecule highly related to cardiovascular processes, by stimulating migration, proliferation, angiogenesis, cell survival, and metabolism. Authors should read articles by Dr. Gomez-Muñoz.

- PAI-1. The authors mention that ceramide stimulates the production/secretion of PAI-1. This point is very interesting and should be discussed. PAI-1 is the rtPA inhibitor, being the non-mechanical treatment for ischemic patients. In addition, different works have observed the variation of the ceramide profile depending on the ischemia and the outcome of the patients. When it comes to cerebrovascular problems, it should be discussed.

- The authors should make a table about the works discussed where the disease/problem is exposed with the ceramide profile obtained.

Minor:

Line 58. "The acylation reaction is done by CerS and different CerSs (Cers1–6)"

It is not by CerS and different CerSs. It is done by CerS activity. So far, 6 different CerS have been described....

- Line 79. De novo synthesis takes place in the cytosolic layer of endoplasmic reticulum and mitochondria involving SPT,

Take place in the ER, however, CerS have been also detected in mitochondria

- Table 1. Grammatical errors such as location and Gcosylceramide that is written twice

I would like to write about all the enzymes implicated in the pathways since all of them are important. 

- Figure 2. change P13K is PI3K, change mitochondria efficiency by function. Acronyms are missing in the legend

Author Response

Pisa, February 24, 2022

Dear Editor,

Please find enclosed the revised manuscript “Ceramides as mediators of oxidative stress and inflammation in cardiometabolic disease” by Gaggini M et al. for publication in IJMS

In the new version, we addressed all reviewer comments and modified the text accordingly. For details, please refer to the point-by-point responses to reviewer attached hereby

We wish to thank the Editor and Reviewers for their constructive appreciation and comments, and we hope that the manuscript in the current form is suitable for publication.

Thanking you for your kind attention and consideration, we look forward to hearing from you.

Yours sincerely,

Dr. Cristina Vassalle

The authors have done a substantial job of improving the article. However, there are still a variety of grammatical errors.

Thank you for your appreciation. We regret there were problems with the English style. For this reason, the paper has been carefully revised by a professional certificated language reviewer to improve the grammar and readability.

Apart from this point, the work has weak points:

Major:
- Line 91. "could be also phosphorylated by the ceramide kinase (CerK), to generate ceramide-1-phosphate (C1P) able to modulate insulin signaling in muscle cells and stimulating the incorporation of glucose by the glucose transporter type 4 Glut4"

C1P is not only a molecule that stimulates metabolism, something that the authors have mentioned referring to a review and not to the original article. C1P is a molecule highly related to cardiovascular processes, by stimulating migration, proliferation, angiogenesis, cell survival, and metabolism. Authors should read articles by Dr. Gomez-Muñoz.

We better deepened the point raised by the reviewer as follow:

“…In the Golgi apparatus, ceramides can be further metabolized to complex sphingolipids (e.g., sphingomyelin, glucosylceramides and gangliosides) and could be also phosphorylated by the ceramide kinase (CerK) to generate ceramide-1-phosphate (C1P), able to modulate insulin signaling in muscle cells and stimulate the incorporation of glucose by the glucose transporter type 4 Glut4 [13, 14]. Interestingly, C1P is highly related to cardiovascular processes by stimulating migration, proliferation, angiogenesis, cell survival, and metabolism [15]..”

- PAI-1. The authors mention that ceramide stimulates the production/secretion of PAI-1. This point is very interesting and should be discussed. PAI-1 is the rtPA inhibitor, being the non-mechanical treatment for ischemic patients. In addition, different works have observed the variation of the ceramide profile depending on the ischemia and the outcome of the patients. When it comes to cerebrovascular problems, it should be discussed.

We better discussed this point as follow: “Fibrinolysis is a physiological event aimed to prevent blood clots and dissolve fibrin clots. Increased levels of plasminogen activator inhibitor-1 (PAI-1), the primary blood inhibitor of tissue plasminogen activator (tPA), are related to the inhibition of plasmin degradation of fibrin, and to a decreased fibrinolytic capacity. In the brain, tPA is reported to regulate blood-brain barrier function, promote neurodegeneration, microglial activation, neurite outgrowth and regeneration [92]. At now, the use of recombinant tissue plasminogen activator recombinant tissue-type plasminogen activator (rTPA) represents the only effective pharmacological treatment for ischemic stroke [93]. Treatment of cultured human astrocytes with N-acetylsphingosine, that increase intracellular ceramide via activation of ceramide synthase or sphingomyelin hydrolysis, induced the release of t-PA and decreased the PAI-1 release [94]. Thus, a better kwnoledge of the connection between ceramides, tPa and PAI-1 may be particulary interesting for the possibility to target ceramides in order to modulate the tPA/PAI-1 relationship. Nonetheless, the variation of the ceramide profile seems to depend on the ischemia and to affect the outcome of the patients. In addition to the increase of total ceramide, experimental data from stroke models suggested that the levels of several long-chain and very-long-chain ceramides in brain cells are significantly increased after ischemic injury [95].The accumulation of specific long-chain ceramides, including Cer(d18:1/16:0), Cer(d18:1/18:0), and Cer(d18:1/20:0) togheter with a sphingomyelin decrease has been observed in the gerbil hippocampus during the reperfusion phase [96]. Other data suggested that Cer(d18:1/14:0) and Cer(d18:1/16:0) values significantly increase and that their ceramide synthase isoform, CerS 5 appear upregulated in neurons after hypoxia/reoxygenation[97]. Importantly, following subarachnoid hemorrhage, patients with high concentrations of Cer(d18:1/18:0) in the cerebrospinal fluid presented poor outcomes [98]. Thus, collectively these results identify a novel role for ceramides in the pathogenesis of ischemia and postischemic processes and in their contribute to the prothrombotic phenotype in ischemia/reperfusion. Further studies are needed to investigate the potential of distinct ceramides and S1P as predictors of prognosis and as therapeutic targets, and elucidate their regulatory reciprocal relationship with t-PA, and its inhibitor PAI-1.

As adipokine, glycoprotein PAI-1 is also able to increase accumulation of ceramides in adipocytes and conversely ceramides can induce PAI-1 expression in adipocytes, revealing a bidirectional interplay between PAI-1 and ceramides in inflammation and metabolism of adipose tissue [99-100]. Accordingly, several studies showed that PAI-1 contributes to the worse of obesity, insulin resistance, type 2 diabetes, and atherothrombosis[101,102].”

- The authors should make a table about the works discussed where the disease/problem is exposed with the ceramide profile obtained.

Table 3 was added reporting ceramides profile evaluated in discussed T2DM and IR studies

Minor:

Line 58. "The acylation reaction is done by CerS and different CerSs (Cers1–6)"

It is not by CerS and different CerSs. It is done by CerS activity. So far, 6 different CerS have been described....

 Checked and changed according to reviewer suggerstions

- Line 79. De novo synthesis takes place in the cytosolic layer of endoplasmic reticulum and mitochondria involving SPT,

Take place in the ER, however, CerS have been also detected in mitochondria

 This important point was further stressed according to reviewer suggestions as follow:

“Important intracellular compartment of ceramide metabolism are mitochondria, which contain a variety of sphingolipids, including ceramides, and have their own subset of sphingolipid-generating and -degrading enzymes [10, 11].”

- Table 1. Grammatical errors such as location and Gcosylceramide that is written twice

Checked and corrected

I would like to write about all the enzymes implicated in the pathways since all of them are important. 

Enzymes relevant for the sphingolipid pathways were added in Table 1

- Figure 2. change P13K is PI3K, change mitochondria efficiency by function. Acronyms are missing in the legend

Checked and corrected according to reviewer suggestions

Round 3

Reviewer 2 Report

Congratulations to the authors for the revision they have made to improve the quality of their work.